
# A Scenario-based Case Study: AI to analyse casualties from landslides in Chittagong Metropolitan Area, Bangladesh

Fahim K. Sufi[1], Edris Alam[2, 3], Abu Reza Md. Towfiqul Islam[4]

[1]Federal Government, Melbourne, Australia, VIC 3000
[2]Business Continuity Management & Integrated Emergency Management, Rabdan Academy, Abu Dhabi, UAE
[3]Department of Geography and Environmental Studies, University of Chittagong, Chittagong-4331, Bangladesh
[4]Department of Disaster Management, Begum Rokeya University, Rangpur 5400, Bangladesh

*Correspondence to*: Fahim K. Sufi (research@fahimsufi.com); Edris Alam (ealam@ra.ac.ae)

**Abstract.** Understanding the complex dynamics of landslides is crucial for disaster planners to make timely and effective
decision that saves lives and reduces the economic impact on society. Using the landslide inventory of Chittagong Metropolitan
Area (CMA), we created a new Artificial Intelligence (AI) based insight system for the town planners and senior disaster
recovery strategists of Chittagong, Bangladesh. Our system generates dynamic AI-based insights for a range of complex
scenarios created from 7 different landslide feature attributes. The users of our system can select a particular kind of scenario
out of the exhaustive list of $1.054 \times 10^{41}$ possible scenario sets and our AI-based system will immediately predict how many
casualties are likely to occur based on the selected kind of scenario. Moreover, an AI-based system shows how landslide
attributes (e.g., rainfall, area of mass, elevation, etc.) correlate with landslide casualty by drawing detailed trend lines
performing both linear and logistic regressions. According to literature and the best of our knowledge, our CMA scenario-
based AI insight system is the first of its kind providing the most comprehensive understanding of landslide scenarios and
associated deaths and damages in CMA. The system was deployed on a wide range of platforms including Android, iOS, and
Windows systems so that it could be easily adapted to strategic disaster planners. The deployed solutions were handed down
to 12 landslide strategists and disaster planners for evaluations whereby 91.67% of users found the solution easy to use,
effective and self-explanatory while using via mobile.

## 1 Introduction

Landslides are natural phenomenon that have an adverse effect on human life as well as economy (Rabby and Li, 2020). For
the purpose of reducing the negative impact of landslide and to have an increased level of disaster preparedness (Alam, 2020),
it is crucial to have a multi-dimensional understanding landslide attributes. The complex nature of landslide dynamics makes
it extremely difficult to understand the impact of a particular type of landslide. Bangladesh is susceptible to a variety of natural
and human-induced hazards including tropical cyclones, floods, droughts, earthquakes, tsunamis, and landslides (Alam, 2020).
Particularly, landslides become recurrent phenomena in the Southeast Bangladesh in recent decades. Therefore, the
Government of Bangladesh (GoB) and its coastal residents have been in reducing resultant deaths from tropical cyclones but



still the landslides have caused over 500 deaths in South-East Bangladesh with the majority in informal settlements in Chittagong and Rangamati districts since 2000. The root causes contributing to vulnerability of three different communities in SE part of Bangladesh; Bengali, Tribal and Rohingya refugees were identified (Ahmed, 2021), and effective local risk governance was also promulgated (Alam and Ray-Bennett, 2021). Studies were also conducted in identifying the root causes

and impacts of landslides using qualitative methods (e.g., interviews and surveys) in Chittagong city and Rangamati district (Sultana, 2020). However, there is further scope to apply Artificial Intelligence (AI) driven techniques to identify physical parameters that significantly influence deaths associated with landslides. As such, in this paper, we deployed a new scenario-based AI insight system, that facilities in-depth understanding of landslide hazard enhances "risk perception" and raising the level of "disaster preparedness" in relation to landslides.

Geo-structural and causative-factor based analysis were applied for exploring landslide susceptibility zoning. Landslide susceptibility and risk assessment have been studied at global levels (Lin et al., 2017; Stanley and Kirschbaum, 2017). Geo-spatial technologies such as the application of Geographical Information System (GIS), Global Positioning System (GPS), and Remote Sensing (RS) have recently taken prominence in the hazard assessment and risk identification to assist in decision making related to landslide disaster risk management (Lissak et al., 2020; Tan et al., 2021). GPS is a space-based navigation

satellite system that acquires information relating to exact location and time in all weather conditions, anywhere in the world and it assists in collecting and storing landslide information. GIS is used in collecting, storing, and analysing geographic information and their non-spatial attributes. A plethora of studies have been conducted using GIS for landslide hazard and risk assessment (Senouci et al., 2021). Remote sensing is the system where information about the earth surface is obtained without direct contact with it. In recent decades, RS is widely applied for identification of landslide area, vulnerability, and risk

mapping (Mohan et al., 2021). Apart from the aforementioned techniques, machine learning algorithms are gaining prominence in enhancing disaster preparedness and response.

There are varieties of methods available to study landslide susceptibility but not limited to: landslide inventory based probabilistic, deterministic, heuristic, and statistical techniques (Guzzetti et al., 1999). The most used landslide inventory-based probabilistic techniques involve; development of the inventory of landslides, Geo-morphological analysis, and

generating the susceptibility maps based on provided parameters (Duman et al., 2005). Deterministic approaches also familiar as quantitative methods involve quantifying factors; physical factors, e.g., soil, rainfall, vegetation, slope variables to generate maps that displays the spatial distribution of input data (Godt et al., 2008, Yilmaz, 2009). Qualitative approach (Heuristic analysis) involves analysing aerial photographs or conducting field surveys to identify intrinsic properties of landform (Sendir and Yilmaz, 2002). Statistical analysis use sample data to identify the relationship between the dependent variable (the presence

or absence of landslides), and the independent variables (landslides triggering/causative factors (Ahmed, 2015).

Artificial intelligence (AI) methods use some of the statistical concepts. These methods are based on assumptions, predetermined algorithms, and output. AI methods or machine learning methods that used for landslide studies include artificial neural network (ANN), Convolutional Neural Network (CNN), Regression, fuzzy based, hybrid, kernel based and tree-based methods (Qi et al., 2021; Sufi and Alsulami, 2021a; Sufi, 2021). These methods are suitable for generating results regardless



of data types (i.e., both discrete and continuous data) and data limitation (i.e., the types and number of conditioning factors). For example, research in (Sufi, 2021; Sufi and Alsulami, 2021a) uses machine learning algorithms to understand the complex dynamics of global landslides which may help strategic decision makers.

Although these studies provide valuable insight into landslide susceptibility as well as the causes and impacts of landslides on the poor in Chittagong, there is a dearth of research that focuses on AI system to analyse casualties from landslides at small

scale. Reducing disaster deaths through AI at national and local levels is aligned with the United Nations' Sendai Framework (2015-2030) for Disaster Risk Global Target A: 'Substantially reduce global disaster mortality by 2030' and Global Target G: 'Substantially increase the availability of and access to multi-hazard early warning systems and disaster risk information and assessments to people by 2030 (UN, 2021). Since Bangladesh is a signatory to the Sendai Framework; it is important that multi-hazard early warning systems and disaster risk information to community level for all hazards are available by the year

75 2030.

In this paper, firstly, we designed and developed a new scenario-based AI insight system that can connect to a landslide database so as to find out unknown insights from landslide data. Secondly, we connected our scenario-based AI insight system, to a dataset containing landslide information and finally, we demonstrated dynamic generation of AI based insights based on specific scenarios. Section 3 (results) shows the positive correlation of Area of Mass as well as rainfall towards the number of

casualties.

Equipped with this AI Insights, a disaster recovery planner and strategist can make informed, timely and evidence-based decisions that can save lives and reduce the economic impact of likely disasters on a society. Moreover, the AI insights would support policy planners to understand the characteristics of landslides in a particular area and provide useful guidance for policy implementation.

**2 Materials & Methods**

First, the data was obtained from, previous landslide catalogue, local histories, archive of institutional and administrative records, newspapers, reports, digital archives, and published peer reviewed journal papers dedicated to landslides in Chittagong Metropolitan Area (CMA) and subsequently cleaned and transformed before modelling. Data collected from secondary sources were validated through field visits and investigation and to identify accurate location of landslide occurrence. Following this,

data modelling using the best practice was performed and then the data was visualized and analysed using AI Systems and algorithms. The details of these AI based analysis is portrayed within this section. Finally, data driven insights were generated. Fig. 1 demonstrates the step-by-step process for generating AI insights on CMA Landslide data. Following subsections describes study area selection, sources of data, preparing the data, modelling the data, visualizing the data, analysing with AI based algorithms (like linear regression, logistic regression, and decomposition tree analysis).




## 2.1 Study Area Selection

Chittagong was selected as the area of study, as the residents of the city have experienced a record of deaths associated with landslide since the year 2000. For example, on the 11th of June 2007, landslide events alone caused the death of 128 casualties and 100 injuries in places adjacent to hilly areas because landslides were triggered by heavy rainfall (610 mm) for eight consecutive days. Five year later, on the 26th of June 2012, another eight days of continuous rainfall (889 mm) triggered landslides, that led to 90 casualties (Alam, 2020). These landslide events occurred in hill cutting areas characterized by high angles/slope. Slope failure in these fragile hilly areas occurs during the rainy reason between the months of June and September.  It is important to note that population in Chittagong has increased four times the number since 1974 compelling a significant number of people living in highly vulnerable areas (Fig. 2).

Chittagong lies along the Western margin of the tectonically active Chittagong-Tripura folded belt. The district is located between 20'35oN and 22'59oN latitude, and 91'27o E to 92'22o E longitude (Fig. 3).  Hills in the district are mainly composed of weathered and loose sedimentary rocks of tertiary (65–1.8 Ma) age which are prone to landslides. The mean monthly maximum and minimum of temperature ranges between 78.76–90.44°F and 55.88–77.38°F and the monthly average minimum of rainfall is 0.66 mm in the month of January and maximum rainfall 74.70 mm in the month of July. The average rainfall per year is about 2794 mm (BMD, 2020). The North-Western and monsoon clouds are primarily responsible for the rainfall in the area and almost 90% of the total yearly precipitation takes place between the months of June and October (BBS, 2011). The total area of Chittagong City corporation is about 170.41 km2. The urban population of Chittagong district was only 0.90 million in 1974 which increased to. 3.15 million in 2011, representing an increase of the urban population by 583% in the last 37 years. The current population density in Chittagong district is 6992 persons/km(sq.), from previously it was 2695 persons/km(sq.) in 2001 (BBS, 2011).

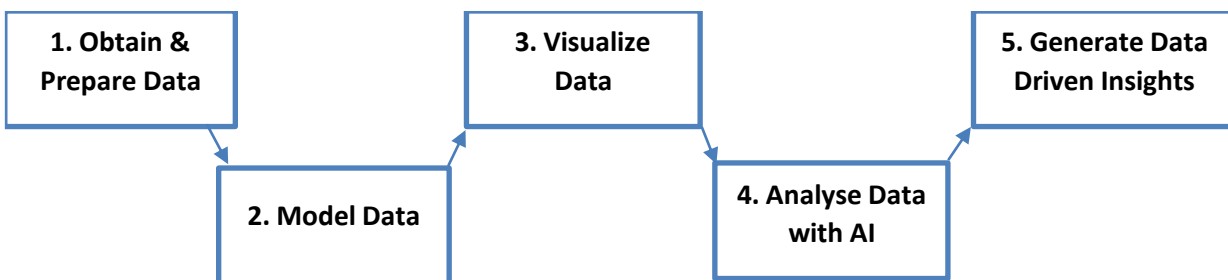

**Figure 1.  High Level Methodology of AI insight system for analysing landslide in CMA location of the SE Bangladesh particularly landslide susceptible areas in Chittagong, Rangamati and Cox's Bazar districts.**

## 2.2 Obtain and Prepare Data

Data can be sourced from one or more sources whereby, these sources can be multiple ranging from online databases, websites, excel files, flat files, Web Based Application Programming Interfaces (APIs) or even PDF files. After identifying the data

sources, the data is obtained with integration tools like SQL Server Integration Services (SSIS), Power BI Query Editor, Oracle
data integrator, Tibco Pervasive Integration etc. These data integration tools facilitate the Export, Transform, Load (ETL)
process which obtains data from many different sources into a data warehouse whereas specialized programming languages
like Mashup (M) language is used for data transformations and data cleansing.

**Figure 2. Landslide vulnerability in different areas of Chittagong (map and photo) (Source: Field visit, October 2018).**

Data transformation and data cleansing can be referred to as "data preparation", since data needs to be first transformed into
right format before the data is modelled or analysed. For our research, we obtained publicly available data directly from PDF
file (Rahman et al., 2016) and then we transformed the data in a suitable format that allows faster analysis. After data
transformation, the feature attributes of CMA landslide data were better understood after date preparation. Table 1 shows the
detailed statistics of the CMA Landslide data. Understanding the statistics for CMA Landslide Feature Attribute details that
are crucial before proceeding to the next steps of the methodology, namely Model Data, Visualize Data, and Analyse Data
with AI.



## 2.3 Model Data

Data modelling is the most important stage within the process of Generating Data Driven Insights and when it is done correctly, the AI driven solution can produce powerful insights with minimum delay. During this stage, relationships among different
sets of data is drawn with the right cardinality.

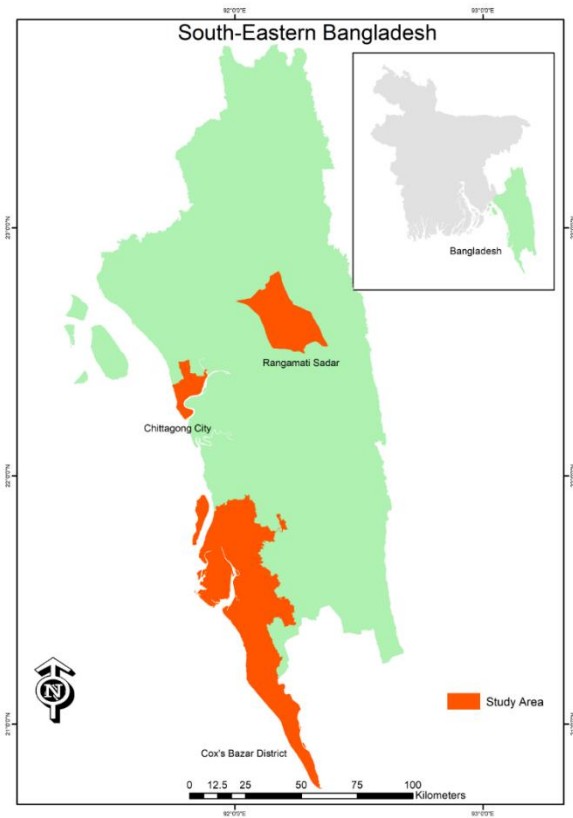

**Figure 3. Location of the SE Bangladesh particularly landslide susceptible areas in Chittagong, Rangamati and Cox's Bazar districts.**

As seen from the Fig. 4, the data obtained for this paper were arranged in a Star Schema (Microsoft, 2022), where the main
factual data resides in the Centre (referred as Landslide DB). Surrounding the fact tables, there are dimension tables that include: Types, State, Date, Hill Name, Style. This arrangement of Star Schema allows controlling the fact table (i.e., Landslide DB) with one-way filtering of information by Types, State, Date, Hill Name as well as Style.  The main benefit of the start schema technique over other data modelling techniques (e.g., flattened table, snowflake etc.), is the speed since it provides more accurate results during data analysis (Ferrai, 2021).





### 2.4 Visualize Data


Once the data model was completed, we use State, Rainfall (mm), Elevation (m) and types information to filter the factual data that drives the AI based insights. A wide range of visualizations like slicer, Bing Maps, Key influencers were used. Changing the values for each of the filters (e.g., State to Dormant or Stabilized), filters the fact table Landslide DB, which in turns changes the key influence's (Fig. 5).

### 2.5 Analyse Data with AI


Within this research, key performance indicator (KPI) visualization was used to analyse casualty (from landslide) and it was explained by following list of feature attributes as named below:

- Area of Mass (m2)
- Elevation (m)
• Hill Name
- Rain fall (mm)
- State
- Style
- Types
• Date

This analysis used machine learning algorithms provided by ML.NET (Sufi, 2021) to figure out what matters the most in driving landslide feature attribute. As seen from Fig. 6, the analysis process uses CMA landslide Data, ranks the factors that matter, contrasts the relative importance of these factors, and displays them as key influencers for both categorical and numeric metrics.

As seen from Fig. 6, two main categories of AI-based statistical analysis are executed on the CMA Landslide data, namely Transformation (Sufi, 2021; Sufi and Alsulami, 2021a), Decompression Analysis (Sufi, 2021; Sufi and Alsulami, 2021a) and Regression Analysis. Transformation analysis is executed for preparing the CMA Landslide data before running the regression analysis and within the transformation, three algorithms are executed, and they include:

One-hot encoding: Calling OneHotEncoding() method within Microsoft.ML.Transforms class results in converting categorical
information into numeric values for efficient and effective processing of machine learning algorithms (Sufi, 2021; Sufi and Alsulami, 2021a).

Replacing missing value: Calling ReplaceMissingValues() method within Microsoft.ML.Transforms class results in replacing the missing value with either default, minimum, maximum, mean, or the most frequent value (Sufi, 2021; Sufi and Alsulami, 2021a).


Normalize mean variance: Calling NormalizeMeanVariance() method within Microsoft.ML.Transforms class results in adjusting values measured on different scales to a notionally common scale with computed mean and variance of the data (Sufi, 2021; Sufi and Alsulami, 2021a).

Once the CMA landslide data is prepared for regression analysis, two different types of regressions are performed. For numerical features, linear regression is performed using Microsoft's ML.Net's SDCA Regression implementation (Sufi, 2021). Linear

Regression is one of the simplest machine learning algorithms that comes under Supervised Learning technique, and it is used for solving regression problems. Moreover, it is used for predicting the continuous dependent variable with the help of independent variables. The goal of the Linear regression is to find the best fit line that can accurately predict the output for the continuous dependent variable by finding the best fit line, algorithm establish the linear relationship between dependent variable and independent variable in the form of $y = b_0 + b_1 x_1 + \varepsilon$. On the other hand, for categorical feature, logistic regression is

performed using ML.Net's L-BFGS logistic regression (Nocedal, 1980) logistic regression is one of the most popular machine learning algorithms that comes under supervised learning techniques since it can be used for classification as well as for regression problems. Logistic regression is used to predict the categorical dependent variable with the help of independent variables using $\log \left[ \frac{y}{1-y} \right] = b_0 + b_1 x_1 + b_2 x_2 + \cdots b_n x_n$. The output of Logistic Regression problem can only be between the 0 and 1. Moreover it can be used where the probabilities between two classes is required, such as whether it will rain today or not, either 0 or 1, true

or false etc.

Other than using linear regression and logistics regression, this study also used decomposition analysis with decomposition tree. Decomposition Tree Visual is a valuable tool for ad hoc exploration and conducting root cause analysis, while allowing the user to visualize the data across multiple filter attributes or dimensions.

Our implementation of decomposition analysis allows the visualization of landslide casualty data over a range of landslide feature

attributes, namely: Area of Mass, Elevation, Rainfall, State and Types. As shown in Fig. 7, interactive root-cause analysis and data exploration were supported by aggregation of data and drill-down, where a user can click and find out what feature attribute causes the highest or lowest casualty.

For feature attributes (i.e., area of mass, elevation, rainfall, state, type, date etc.), $T = \{T^1, T^2, T^3, \dots, T^N\}$, where N is the number of total filter attributes within a data-set (i.e., the cardinality of T, $|T| = N$ ), each feature attribute can form one or

many filtered conditions, as follows:

$$T^1 = \{T_1^1, T_2^1, T_3^1, \dots, T_P^1\}, such\ that\ |T^1| = P \tag{1}$$

$$T^2 = \{T_1^2, T_2^2, T_3^2, \dots, T_Q^2\}, such\ that\ |T^2| = Q \tag{2}$$

$$T^3 = \{T_1^3, T_2^3, T_3^3, \dots, T_U^3\}, such\ that\ |T^3| = U \tag{3}$$

$$T^N = \{T_1^N, T_2^N, T_3^N, \dots, T_N^3\}, such\ that\ |T^3| = V \tag{4}$$

Each of these filter conditions can filter r number of rows, $r \in \{1, 2, 3, \dots R\}$ from the dataset. Proceeding with this context, we defined Casualty from landslide as Eq. (5).

$$C_i^n = \sum_{i=0}^{r} (casulty\_count),\ Where, r\ is\ the\ rows\ effected\ by\ filter\ attribute\ condition\ T_i^n \tag{5}$$





Our decomposition tree visualization (supported by AI) allows the user to find the next filter attribute condition to drill down into based on either high or low values.


*High Value*: This mode considers all available filter attribute conditions and determines which one to drill into to obtain the highest value of the measure being analysed. Therefore, the high-value AI split mode finds the most influential filter attribute condition $T_i^n$, for which the highest level of casualty occurs, is represented by

$$\exists T_i^n \subseteq T \mid C_i^n > C_j^m, \forall n, m \subseteq \{1,2,3, \dots, N\} \land \forall i, j \subseteq \{1,2,3, \dots\} \tag{6}$$

*Low Value*: This mode considers all available filter attribute conditions and determines which one to drill into to obtain the lowest value of the measure being analysed. Therefore, the low-value AI split mode finds the most influential filter attribute condition $T_i^n$, for which the lowest level of casualty occurs, is represented by

$$\exists T_i^n \subseteq T \mid C_i^n < C_j^m, \forall n, m \subseteq \{1,2,3, \dots, N\} \land \forall i, j \subseteq \{1,2,3, \dots\} \tag{7}$$

In this way, the AI Split allows the user to understand the details of the root cause. This AI split based decomposition analysis 220 was used in our most recent study on knowledge discovery on global landslides.

**TABLE 1**

**LANDSLIDE ATTRIBUTE, DATA TYPE & DATA DISTRIBUTION**

| Type of Attribute | Data Type | Attribute Distribution | Other Attribute Details |
|---|---|---|---|
| **ID** | Integer | 57 distinct, 57 unique | 57 Distinct, 57 Unique<br>Value Example: Ranges from 1 to 57 |
| **Latitude** | Decimal | 50 distinct, 44 unique | 50 Distinct, 44 Unique |
| **Longitude** | Decimal | 54 distinct, 51 unique | 54 Distinct, 51 Unique |
| **Elevation** | Decimal | 56 distinct, 55 unique | 56 Distinct, 55 Unique |
| **Date** | Date<br>(dd-mm-yyyy) | 6 distinct, 0 unique | 6 Distinct, 0 Unique, **34 Empty**<br>● Valid 40%<br>● Error 0%<br>● Empty 60% |



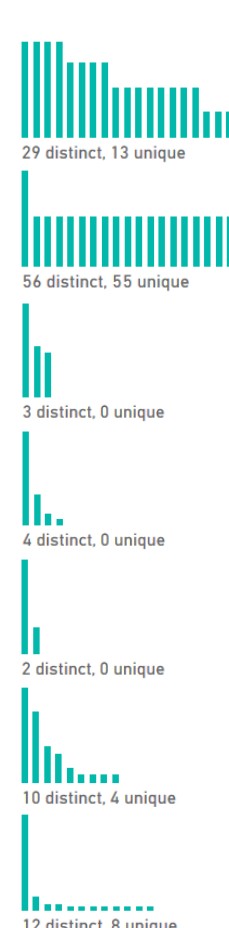

| Field | Type | Statistics |
|---|---|---|
| **Hill Name** | Text | 29 Distinct, 13 Unique<br>Value Example: *Lebu Bagan, Ctg. University, Foy'z Lake Zoo hill, Medical Hell, Tankir Pahar, Sekandar Para …* |
| **Area of Mass** | Decimal | 56 Distinct, 55 Unique |
| **Types** | Text | 3 Distinct, 0 Unique<br>Value Example: *Slide, Fall, Topple* |
| **State** | Text | 4 Distinct, 0 Unique<br>Value Example: *Active, Stabilized, Dormant, Reactivated* |
| **Style** | Text | 2 Distinct, 0 Unique<br>Value Example: *Single, Successive* |
| **Rainfall** | Integer | 10 Distinct, 4 Unique, **18 Empty**<br>● Valid 68%<br>● Error 0%<br>● Empty 32% |
| **Casualty** | Integer | 12 Distinct, 8 Unique |



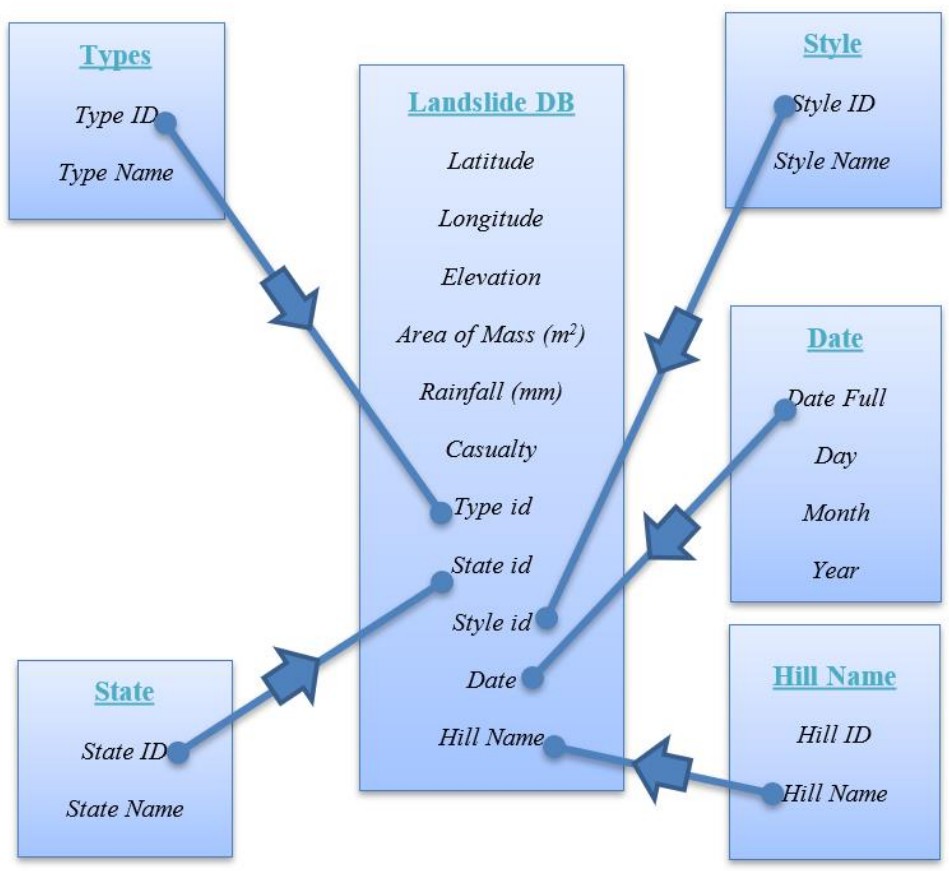

**Figure 4. Data Modelling of CMA Landslide Database.**



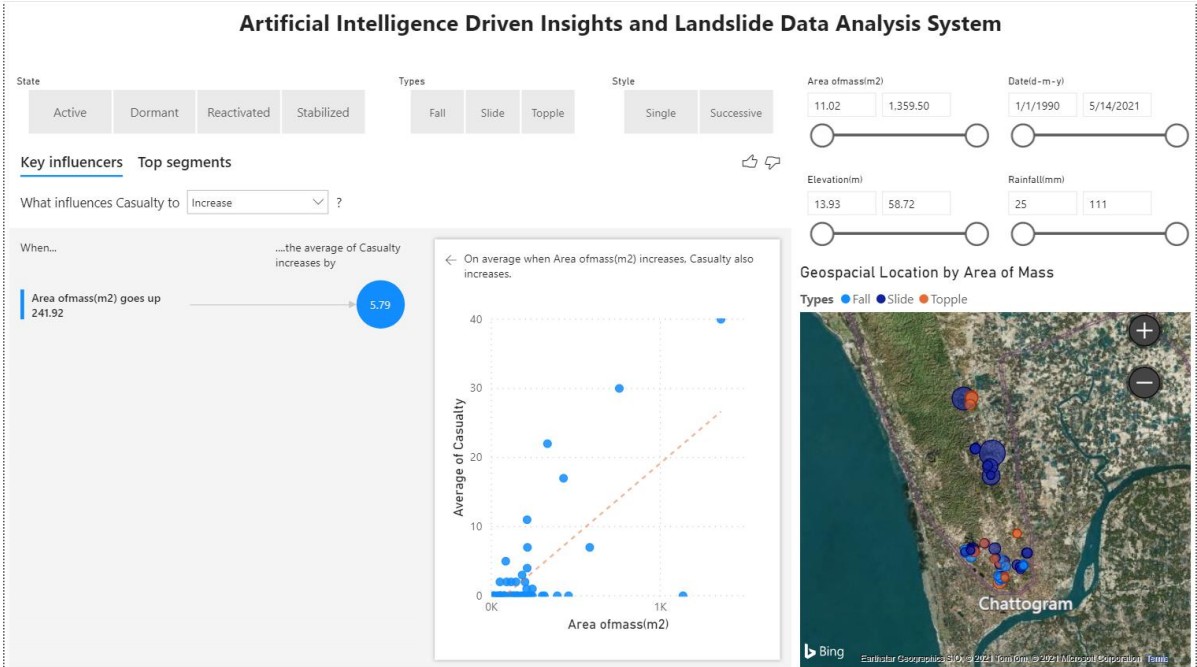

**Figure 5. AI Based insight and landslide analysis system.**




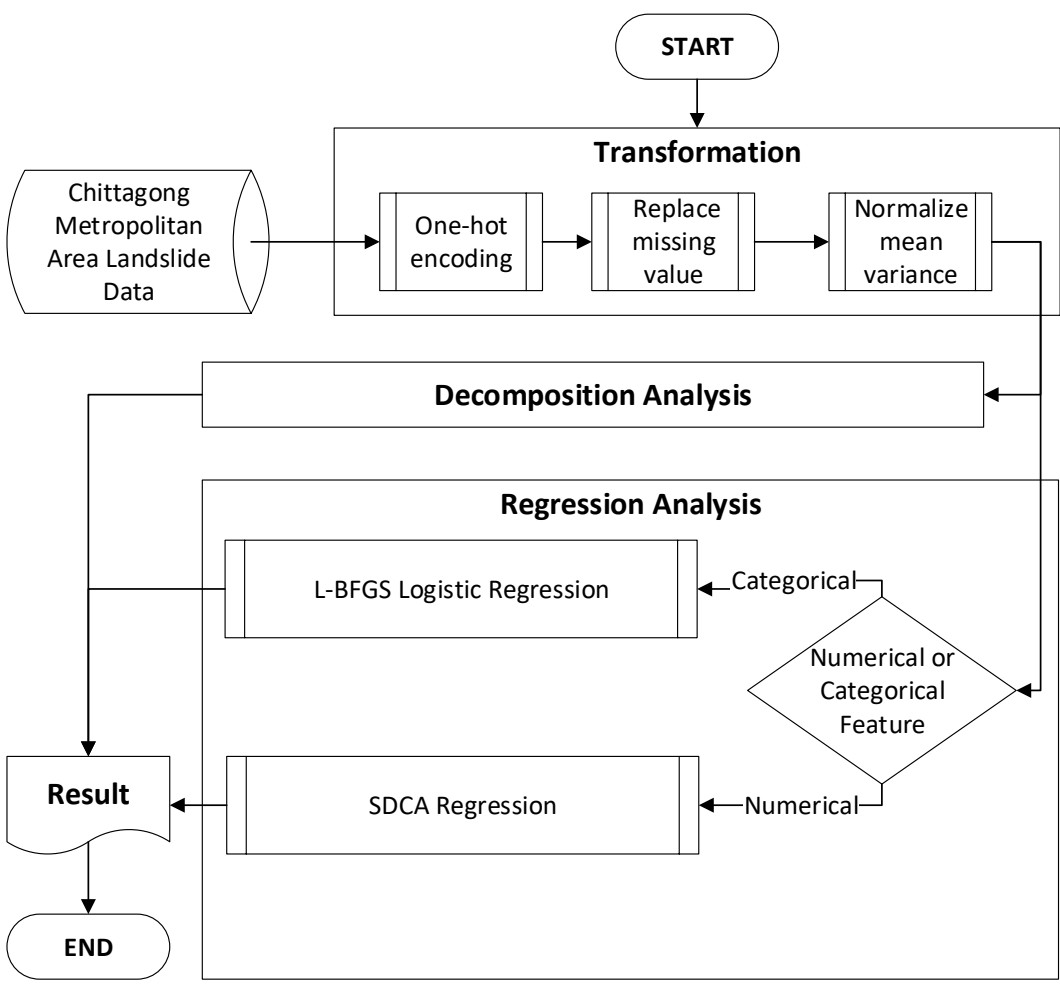

**Figure 6. The Process of obtaining AI Insights from CMA Landslide data using Machine Learning Algorithms.**




## 2.6 Generate Data Driven Insights

Within this phase, valuable insights are produced. The success of this phase depends on the success of previous activities such as preparing (i.e., transformation and cleaning) of data, selection of the right AI Visual and most importantly data modelling. Following the employment of the AI based Key Influencer Visualization, the key factor that influence the number of casualties

was *Area of Mass (m²)*. The other factor that influences the casualty under specific condition was *Elevation (m)*. Data driven insights are generated by configuring one or more scenarios. A scenario can easily be created using our system either by clicking the desired buttons shown in Fig. 8 or by changing the sliders of Fig. 8. In our system, we have created Scenario (*S*)

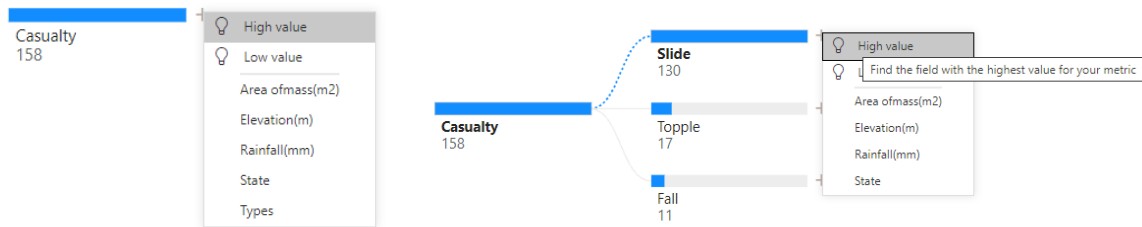

**Figure 7.  Decomposition tree visualization allows the user to perform interactive analysis by Area of Mass, Elevation, Rainfall, State and Types.**

from 7 different attributes namely, Types ($T^1$), State ($T^2$), Style ($T^3$), Elevation ($T^4$), Area of Mass ($T^5$), Rainfall ($T^6$) and Date ($T^7$) as parameters. Therefore,

$S=[x, y, z, m, n, p, q \mid x \subseteq T^1, y \subseteq T^2, z \subseteq T^3, m \subseteq T^4, n \subseteq T^5, p \subseteq T^6, q \subseteq T^7]$          (8)

$T^1 = \{Slide,\ Fall,\ Topple\}$          (9)

$T^2 = \{Active,\ Stabilized,\ Dormant,\ Reactivated\}$          (10)

$T^3 = \{Single,\ Successive\}$          (11)

$T^4 = \{$ *13.93, 15.11, 15.93, 18.1, 18.1, 19.33, 19.84, 21.31, 21.59, 22.64, 23.12, 23.5, 24.71, 26.57, 26.98, 27, 28.41, 29.28,*

*30.82, 31.66, 32.39, 32.44, 32.56, 34.21, 34.63, 35, 35.18, 36.68, 37.54, 37.64, 37.92, 38.51, 38.64, 39.81, 40.19, 40.68, 41.18,*

*41.22, 44.26, 44.46, 45.12, 45.36, 45.42, 45.69, 46.07, 46.4, 46.51, 47.04, 48.36, 48.51, 48.67, 50.12, 51.79, 55.03, 55.95,*

*56.36, 58.72}*

          (12)

$T^5 = \{$ *11.02, 15.03, 16.5, 31.67, 33, 45.86, 47.04, 50.17, 50.26, 52.3, 56.05, 59.1, 71.93, 71.93, 75.88, 76.43, 77.81, 84.56,*

*89.91, 105.38, 116.32, 118.34, 126.7, 130.32, 136, 145.06, 145.5, 152.79, 153.55, 157.07, 175.81, 181.7, 184.13, 188.59,*

*191.64, 198.89, 208.57, 209.12, 211.06, 211.61, 212.7, 213.26, 226.23, 232.52, 233.06, 241.79, 242.53, 301.06, 313.42,*

*331.84, 390.34, 427.04, 456.7, 582.27, 757.61, 1134.77, 1359.5}*

          (13)

$T^6 = \{Ø, 25, 26, 46, 50, 54, 55, 77, 88, 111\}$          (14)





$T^7 = \{\emptyset, 11/6/2007, 1/1/1990, 1/7/2011, 3/8/2005, 5/14/2021\}$ (15)

Eq. [14] and Eq. [15] contains null values represented by $\emptyset$.

To calculate the number of possible scenarios, we first need to calculate the possible filter options for each of the feature attributes. For example, as it appears from Eq. (8), Type attribute could have the following filter options:

1. {}
2. {*Fall*}
3. {*Topple*}
4. {*Slide, Fall*}
5. {*Fall, Topple*}
6. {*Slide, Topple*}
7. {*Slide, Fall, Topple*}

Therefore, for Type attribute, there could be 7 possible filter settings as represented by $(2^{|T^1|} - 1)$, which the formula to calculate power set of Type attribute minus 1 (i.e. $P(T^1) - 1$). 1 is deducted since the power set also includes empty set and selection of empty set is not a supported option by the system presented.

Hence, the total number of possible scenarios could be calculated as,

$|S| = (2^{|T^1|} - 1)X(2^{|T^2|} - 1)X(2^{|T^3|} - 1)X(2^{|T^4|} - 1)X(2^{|T^5|} - 1)X(2^{|T^6|} - 1)X(2^{|T^7|} - 1) = 1.054X10^{41}$ (16)

The purpose of this section is not only to produce an exhaustive list of insights from the landslide data, but also to demonstrate the ability of the designed AI solution for producing insights on any scenarios out of the $1.054X10^{41}$ possible scenarios (as shown in Eq. (16)). In the next section we will explore results (i.e., AI insights obtained from few of these scenarios).

## 3. Results

Since this study was reported on two different AI based techniques namely: automated regression analysis and decomposition analysis. Therefore, within result sections we will briefly describe AI insights derived from both methodologies. Table 2 demonstrates the outcome of conducting regression analysis on various scenarios. In Table 2, the three columns represent AI based insight, the result obtained through the system interface and the scenario condition. Seven rows of Table 2 represent five different scenarios (row 3 & row 4 represent same scenario and row 5 & row 6 represent another single scenario). For example,
Row 5 and Row 7 of Table 2 has the following scenario condition (i.e., both belongs to the same scenario):

1. State= *"Stabilized"*,
2. Type= *All*,
3. Style= *All*,
4. Area of Mass= *All*,
5. Date= *All*,
6. Elevation= *{p|29.05≤ p ≤58.72}*,



7. Rainfall = $\{n\,|\,43 \leq n \leq 111\}$

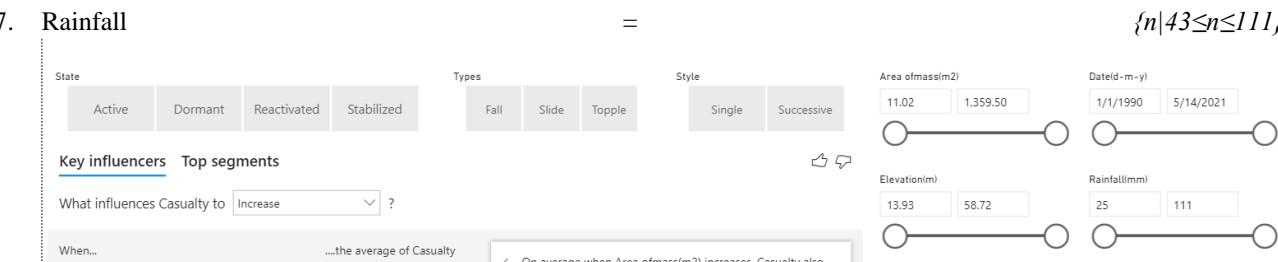

**Figure 8. Filter Area for selection of landslide attributes.**


**TABLE 2 Insights generated by the proposed systems on 5 different scenarios** (Copyright Statements for Microsoft Bing: Earthstar Geographics SIO, © 2022 TomTom, © Microsoft Corporation, © OpenStreeetMap)

| AI INSIGHTS GENERATED ON SPECIFIC SCENARIOS AI Insight | AI Based System Settings | Scenario |
|---|---|---|
| 1. When Area of Mass Goes up 241.92 -> The average of Causality increases by 5.79 | | State= *All*, Type= *All*, Style= *All*, Area of Mass= *All*, Date= *All*, Elevation= *All*, Rainfall = *All* |
| 2. When Area of mass (m2) goes up 539.49->the average of Causality increases by 15.97 | | State= *"Dormant"*, Type= *All*, Style= *All*, Area of Mass= *All*, Date= *All*, Elevation= *All*, Rainfall = *All* |




| AI INSIGHTS GENERATED ON SPECIFIC SCENARIOSAI Insight | AI Based System Settings | Scenario |
|---|---|---|
| 3. When Area of mass (m2) goes up 137.08->the average of Causality increases by 1.53 | | State= *"Stabilized"*, Type= *All*, Style= *All*, Area of Mass= *All*, Date= *All*, Elevation= *All*, Rainfall = *All* |
| 4. When rainfall (mm) goes up 29.29->the average of Causality increases by 0.49 | | State= *"Stabilized"*, Type= *All*, Style= *All*, Area of Mass= *All*, Date= *All*, Elevation= *All*, Rainfall = *All* |
| 5. When Area of Mass (m2) goes up 71.31, the average of Casualty increases by 0.69 | | State= *"Stabilized"*, Type= *All*, Style= *All*, Area of Mass= *All*, Date= *All*, Elevation= *{p/29.05≤ p ≤58.72}*, Rainfall = *{n/43≤n≤111}* |



| AI INSIGHTS GENERATED ON SPECIFIC SCENARIOSAI Insight | AI Based System Settings | Scenario |
|---|---|---|
| 6. When Elevation (m) goes up 5.62, the average of Casualty increases by 0.6 | | State= *"Stabilized"*, Type= *All*, Style= *All*, Area of Mass= *All*, Date= *All*, Elevation= *{p/29.05≤ p ≤58.72}*, Rainfall = *{n/43≤n≤111}* |
| 7. When, Area of mass goes up 149.35, the average of Casualty increases by 1.76 | | State= *"Stabilized"*, Type= *"Slide"*, Style= *"Single"*, Area of Mass= *All*, Date= *All*, Elevation= *{p/18.25≤ p ≤58.72}*, Rainfall = *{n/24≤n≤105}* |

Once the above scenario was configured using the software interface (as shown previously in Fig. 8), the AI insight dynamically executed the regression analysis and described the following insights in plain English language:

  1) *When Area of Mass (m2) goes up 71.31, the average of Casualty increases by 0.69*
  2) *When Elevation (m) goes up 5.62, the average of Casualty increases by 0.6*

In other words, for the selected scenario casualty is positively correlated with both Elevation and Area of Mass. The system dynamically calculated the coefficients of the positive correlation as soon the user configured the scenario. Hence, the user of the system doesn't need to know the complexity of ML algorithms, or the user doesn't need to understand when to use linear

regression and when to use logistic regression. The proposed interactive system executes the right regression depending on the configured scenario of the user. A strategic decision maker can obtain the AI insight in plain English and take appropriate decision based on the AI Insight.



The purpose of this section is not just to generate an exhaustive list of AI insights for all $1.054 X 10^{41}$ possible scenarios. The rest of the table 2, demonstrate some other AI insights generated by 4 other scenarios to demonstrate the applicability of the

system.

Both Fig. 9 and Fig. 10 shows insights generated through decomposition analysis. Firstly, in Fig. 9 a user selected the entire range of data using the option box and sliders at the top of the figure. Then, the user selects "High Value" (as shown previously in Fig. 7) to find out what causes the highest number of casualties. Immediately after selection, the system shows to the user that when *Type* is "slide" casualty is highest. The system also provides visual cues to the user showing *Type=Slide* caused 130

casualties out of total 158 Casualties. Hence, the user can confidently perform root cause analysis without any knowledge about underlying statistical methods. Furthermore, the user can select "High Value" again (as shown previously in Fig. 7), and find out that when rainfall is 88, number of casualties is at its peak (i.e., Fig. 9 shows when rainfall is 88, there were 98 casualty).  Similarly, the user can continue drilling down into further root-causes to find out all the features and corresponding feature values that caused the highest number of casualties being recorded.  Follow on from condition like "type is slide",

"rainfall is 88", Fig 9 shows the other feature conditions that caused the highest number of casualties, namely: "state is Dormant", "Area of mass is 1269.5", and "elevation is 46.07". Therefore, Fig. 9 shows an interactive tool for discovering hidden insight into what caused the highest number of casualties.

Using the decomposition tree visual, the user can also find out what causes the lowest number of casualties and to find out how a particular feature effect the number of casualties. For example, Fig. 10 shows what caused the highest casualties when

*Types*= "Topple". As depicted from Fig. 9 the highest number of casualty (for *Types*= "Topple") was found to be *State= Active* (i.e., most important factor) and *Area of Mass= 427.04 (m²)* (i.e., second most important factor), *Elevation=31.66 (m)* (i.e., third most important factor), *Rainfall = 55 (mm)* (i.e., last important factor).

It is crucial to highlight the fact that the proposed system is robust enough to provide critical insights from the underlying data on any number of scenarios as shown in Table 2, Fig. 9 and Fig. 10 using regression analysis and decompression analysis.




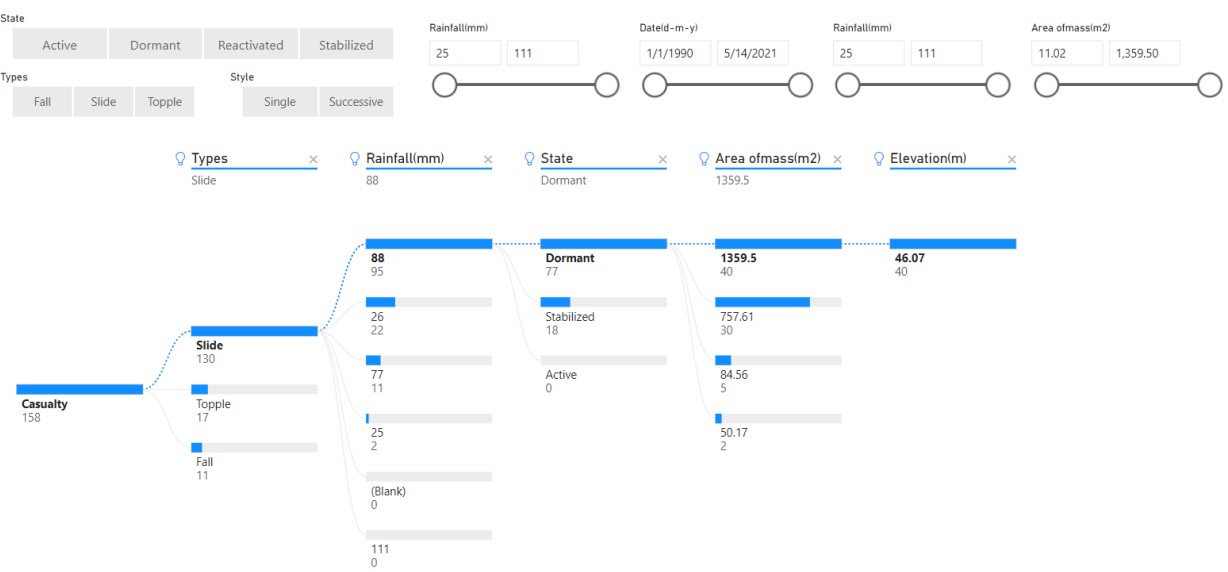

**Figure 9. Decomposition analysis showing what causes the most casualties.**


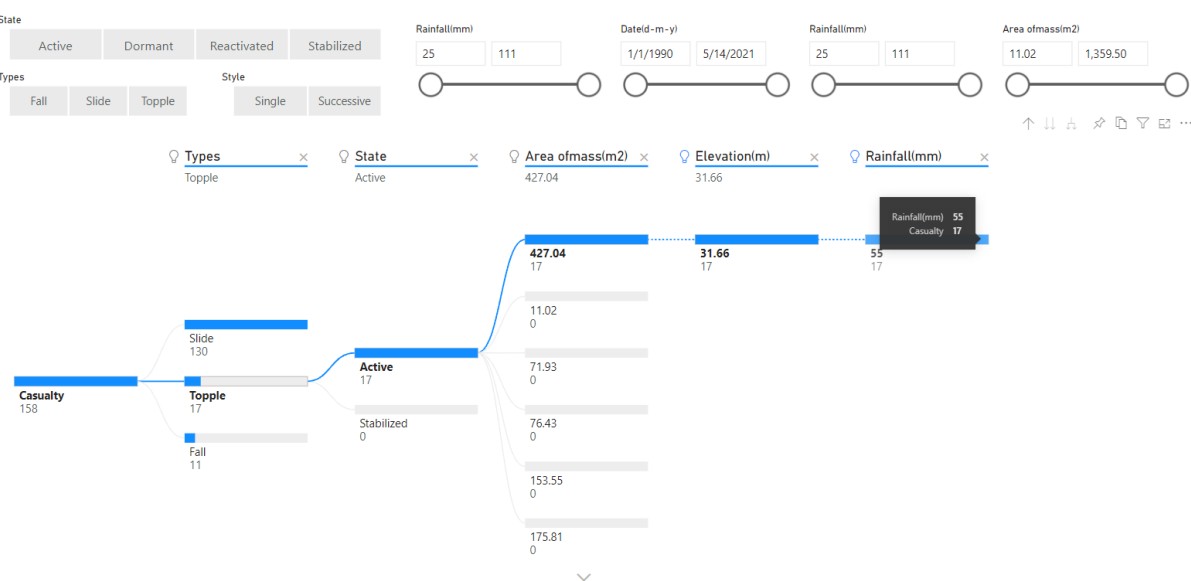

**Figure 10. Decompression analysis showing what caused the highest casualties when Types= "Topple".**



## 4. Discussion

Since all the existing studies in landslide research do not support mobile app based AI insight (Ahmed, 2015; Ahmed, 2021; Alam, 2020; Alam and Ray-Bennett, 2021; Guzzetti et al., 1999; Lin et al., 2017; Lissak et al., 2020; Mohan et al., 2021; Qi et al., 2021; Rabby and Li, 2020; Rahman et al., 2016; Sendir and Yilmaz, 2002; Senouci et al., 2021; Stanley and Kirschbaum, 335 2017; Sultana, 2020; Tan et al., 2021; Yilmaz, 2009), it is not possible to obtain instant insights by a strategic decision maker, if he or she is only equipped with mobile phone. In this study we have deployed the proposed solution in desktop, tablet and even mobile environment since the strategic decision maker can be eager to find out AI based insights being remotely located at a possible landslide incident. As shown in Fig. 11, the AI based auto-regression is being executed in Samsung Note 10 mobile phone. Fig. 12 shows the decomposition analysis on the user's selected scenario is being executed in mobile 340 environment as well. Fig. 13 demonstrates the solution deployed through iOS App in Apple iPad 9th Generation running iOS version 15.1. Fig. 14 showcases the deployed Android app running in Samsung Galaxy Tab A7 running Android 11.

To test, assess, and evaluate the proposed AI based landslide analysis system, the fully deployed solutions were given to 12 landslide researchers, disaster strategists and town planners. The users were primarily located in the following area using their GPS enabled devices for obtaining location centric insights using the proposed solution:

Colony para, the University of Chittagong

Motijharna, Chittagong City

Matiranga, Rangamati

Table 3 shows the platform and user details for these tests and evaluations. As seen from Table 3, the proposed solution was tested on a wide range of devices including both mobile and tablet. Since strategic decision makers often makes their decision 350 on the site of the landslide or away from offices, they need mobile solutions deployed on tablets and mobiles through iOS or Android apps. After the completion of the test and evaluation, detailed feedback regarding usability and appropriateness of the deployed solutions were obtained via Microsoft Form based questionnaires (i.e., office 365 cloud-hosted). 11 out of the 12 users (i.e., 91.67%) found the solution easy to use, effective, and self-explanatory. However, one user preferred using the solution in his desktop computer through the cloud-based interface.

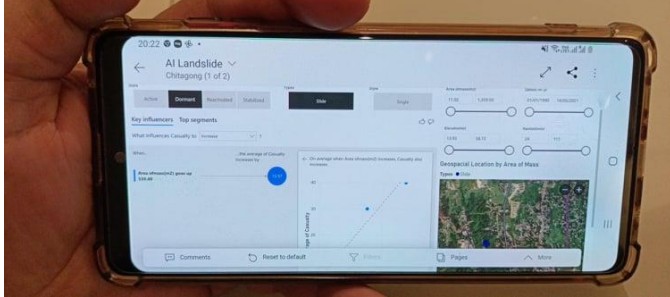


**Figure 11.  The proposed system running regression analysis on Samsung Note 10 mobile device and providing AI based insights on CMA landslides** (Copyright Statements for Microsoft Bing: Earthstar Geographics SIO, © 2022 TomTom, © Microsoft Corporation, © OpenStreeetMap)**.**




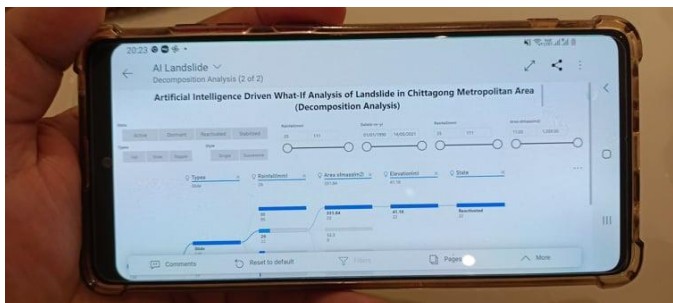

**Figure 12. The proposed system running decompression tree analysis on Samsung Note 10 mobile device and providing AI based insights on CMA landslides** (Copyright Statements for Microsoft Bing: Earthstar Geographics SIO, © 2022 TomTom, © Microsoft Corporation, © OpenStreeetMap).

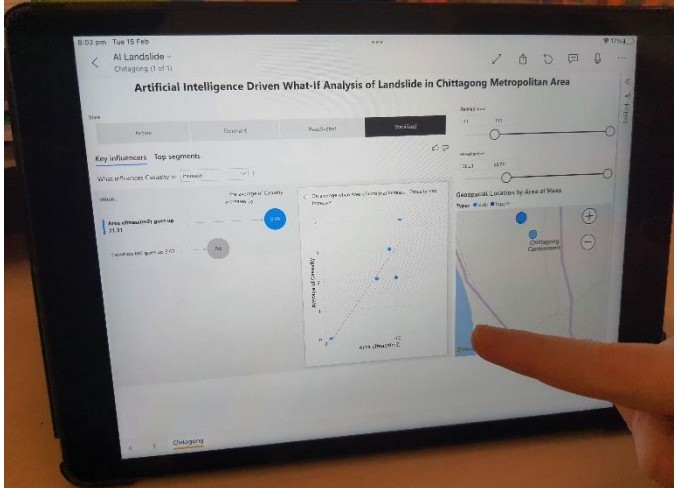

**Figure 13. The proposed system running linear regression on Apple iPad 9th Generation (iOS 15.1)** (Copyright Statements for
Microsoft Bing: Earthstar Geographics SIO, © 2022 TomTom, © Microsoft Corporation, © OpenStreeetMap).

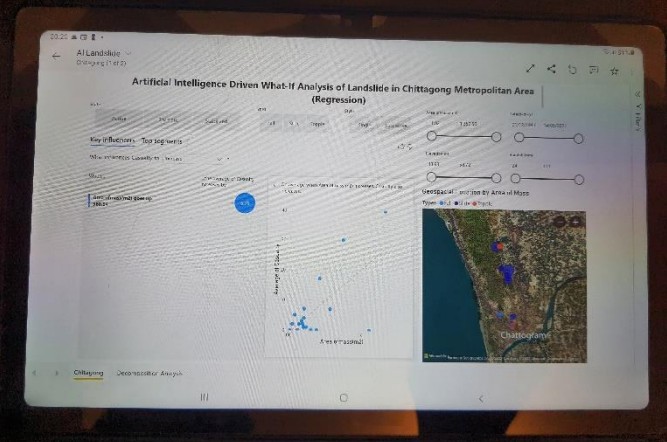

**Figure 14. The proposed system running linear regression on Samsung Galaxy Tab A7 (Android 11)** (Copyright Statements for Microsoft Bing: Earthstar Geographics SIO, © 2022 TomTom, © Microsoft Corporation, © OpenStreeetMap).





**TABLE 3. DETAILS FOR DEPLOYMENT PLATFORM & USERS**

| Number of Users | Device Name | OS Version |
|---|---|---|
| 2 | Samsung Note 10 Lite (Mobile) | Android 11 |
| 1 | Samsung Note 10 Lite (Mobile) | Android 12 |
| 2 | Samsung Galaxy Tab A7 (Tablet) | Android 11 |
| 2 | iPhone 13 (Mobile) | iOS 15 |
| 1 | iPhone 12 (Mobile) | iOS 14 |
| 2 | iPad 9th Generation (Tablet) | iOS 15.2 |
| 2 | iPad Mini 6 (Tablet) | iOS 15 |

Hence, this mobile based AI-Insight system provides a robust and innovative solution for the strategic decision maker who doesn't need to depend on a data scientist to conduct data modelling to obtain valuable insight. By interacting with the proposed system, a strategic decision maker can harness powerful ML algorithms automatically and obtain useful insights.

## 5. User Notes

The ML-based knowledge-discovery solution presented in this study was implemented using Microsoft Power BI, which is freely available for download from https://app.powerbi.com/. The user can download the complete source files (.pbix), along with the CMA landslide data (.csv) files from Author's GitHub site. After downloading and opening the entire solution using MS Power BI Desktop, the user can host the solution to either Microsoft Cloud or within a local network for making it available to other researchers or strategic planners.

The typical users of this system are strategic disaster planners, disaster risk assessor, policymakers, and disaster strategists who are concerned with landslide or landfalls and their subtle impact on society, groups, and locations. This system would allow users to understand the characteristics of global events in a particular area since it provides useful guidance for policy implementation.

## 6. Conclusion

This paper provides a detailed methodological framework for generating AI-based insights on landslides in CMA. This experimentation was performed on a limited dataset containing only 57 records. Sadly, there were several limitations due to the relatively small dataset in terms of empty values within date and rainfall attributes. As evident from Table 1, Date attribute has 34 empty values (i.e., 40% valid and 60% empty values) and rainfall has 18 empty values (i.e., 68% valid and 32% empty).





Despite these limitations on available information, the AI-based techniques like automated regressions (both linear and logistic) as well as decomposition algorithm successfully derived useful insights for the strategic decision-maker.

In the future, we will endeavor to work with more records on landslides outside of CMA regions. Using these large-scale records, we hope to deploy more sophisticated AI-based techniques like Convolution Neural Network (CNN) based deep learning to generate useful insights (since our recent study in (Sufi and Khalil, 2022) has demonstrated that applying CNN on

disaster monitoring harnesses better results). Other than CNN, we also want to use sophisticated AI based techniques as demonstrated in our recent and past studies (Sufi, 2021; Sufi, 2022; Sufi and Alsulami, 2021a; Sufi and Alsulami, 2021b; Sufi and Alsulami, 2022; Sufi and Khalil, 2010; Sufi and Khalil, 2022).

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
