# Peer review of "A Scenario-based Case Study: AI to analyse casualties from landslides in Chittagong Metropolitan Area, Bangladesh"

_Natural Hazards and Earth System Sciences, 2022_

## Author Comment (AC8)

Many thanks to the anonymous reviewer for finding our solution and study innovative. Indeed, this system presents a new method for autonomously extracting AI-driven insights interactively from landslide related data using

Regressions and Decomposition Analysis. This innovative methodology is now being used in other areas of research like cyclones and other natural disasters as evident from the recent citations of this preprint discussion.

We appreciated the interest of the reviewer in our approach with three highly legitimate and relevant queries. Our responses with the corresponding queries are briefed below:

**Query 1:** If other researchers want to apply this methodological framework in a different location, what are the main characteristics defining valuable data helpful in performing meaningful insights?

It is possible to use the methodology explained in this research to apply on landslides (or even other disasters like

Cyclone or Tornado) that happened in other locations. For example, the process of using the same methodology in

Tornado related casualty is explained in our following recent publication:

• Fahim Sufi, Edris Alam, Musleh Alsulami, "A New Decision Support System for Analyzing Factors of

Tornado Related Deaths in **Bangladesh**", Sustainability, Vol. 14, No. 10, 2022 (Impact Factor 3.889).

Similarly, the same method applied in critically analyzing **Australian** cyclones is explained in our following recent publication:

• Fahim Sufi, Edris Alam, Musleh Alsulami, "Automated Analysis of Australian Tropical Cyclones with

Regression, Clustering and Convolutional Neural Network", Sustainability, Vol. 14, No. 16, 2022 (Impact

Factor 3.889).

Moreover, this method could also be used to monitor disasters from any global locations as demonstrated in following publication:

• Fahim Sufi and Ibrahim Khalil, "Automated Disaster Monitoring From Social Media Posts Using AI-Based

Location Intelligence and Sentiment Analysis," in IEEE Transactions on Computational Social Systems, doi:

10.1109/TCSS.2022.3157142, 2022 (https://ieeexplore.ieee.org/document/9737676, Impact Factor 4.747)

As it becomes apparent from these recent publications, the dataset is first required to be cleansed and transformed.

Then, the Microsoft Power BI's Key Influencer visualization is used to analyze the outcome variable (e.g., Casualty)

with respect to a list of available "explain by" variables (e.g., Elevation, Rainfall, Area of Mass, Longitude, Latitude,

Number of Injuries, Style, Types etc.). The detailed process in using Microsoft Power BI's Key influencer visualization is explained at https://learn.microsoft.com/en-us/power-bi/visuals/power-bi-visualization- influencers?tabs=powerbi-desktop.

**Query 2:** What considerations were made to select the collection of feature attributes used to analyze casualties?

Machine Learning (ML) based feature analysis (e.g., linear Regression or logistic Regression) depends on the availability of many feature attributes for understanding their correlations to the outcome variable. In this study,

Casualty was deemed as an outcome variable, since strategic decision makers are always keen on saving precious lives resulting from landslides. Within our dataset, we only had few available features to analyze (e.g., Latitude,

Longitude, Elevation, Area of Mass, Rainfall etc.). After applying our innovative method, our solution found a positive correlation of casualty with "Area of Mass" (as shown in Fig. 5, Row 1 of Table 2, Row 2 of Table 2, Row 3 of Table 2,

Row 4 of Table 2, Row 5 of Table 2, Row 6 of Table 2, Row 7 of Table 2), Rainfall (as shown in Row 3 of Table 2, Row 4

of Table 2), and Elevation (as shown in Row 5 of Table 2, Row 6 of Table 2). Even though we utilized all the available features present within our dataset to obtain relationships with the observed variable (i.e., casualty), we considered appropriate data cleansing prior to the automated ML process. As result of the cleansing process, Elevation and Area of Mass turned out to be decimal type of data and Rainfall turned out to be integer type of data.

Pre-processing the available dataset with appropriate data cleansing and transformation is the key for obtaining better AI-driven insight on casualty.

**Query 3:** Do the selected KPI follow any variables contributing to exacerbating the disaster condition in past landslide-related events?

This study didn't use any KPIs to report the past landslide related casualty. The first sentence in "Section 2.5 Analsys

Data with AI" mentioned that "Key Performance Indicator (KPI) visualization was used to analyse casualty...". In fact, it should be rewritten as "Microsoft Power BI's Key Influencer visualization was used to analyse casualty...". As seen from reference (https://learn.microsoft.com/en-us/power-bi/visuals/power-bi-visualization- influencers?tabs=powerbi-desktop), Key Influencers visualization finds out all the dependent variables along with their relationships to an observed variable. In this study, this Key Influencer visualization found out that "Area of

Mass", "Rainfall", and "Elevation" are three most related feature attributes that have direct correlation with past landslide related Casualties.